# Rapid Weight Gain, Infant Feeding Practices, and Subsequent Body Mass Index Trajectories: The CALINA Study

**DOI:** 10.3390/nu12103178

**Published:** 2020-10-17

**Authors:** Paloma Flores-Barrantes, Isabel Iguacel, Iris Iglesia-Altaba, Luis A. Moreno, Gerardo Rodríguez

**Affiliations:** 1GENUD (Growth, Exercise, NUtrition, and Development) Research Group, Faculty of Health Sciences, University of Zaragoza, Spain Edificio del SAI, C/Pedro Cerbuna s/n, 50009 Zaragoza, Spain; iguacel@unizar.es (I.I.); iglesia@unizar.es (I.I.-A.); lmoreno@unizar.es (L.A.M.); gerard@unizar.es (G.R.); 2Instituto Agroalimentario de Aragón (IA2), 50013 Zaragoza, Spain; 3Instituto de Investigación Sanitaria Aragón (IIS Aragón), 50009 Zaragoza, Spain; 4Centro de Investigación Biomédica en Red de Fisiopatología de la Obesidad y Nutrición (CIBERObn), 28029 Madrid, Spain; 5Red de Salud Materno Infantil y del Desarrollo (SAMID), RETICS ISCIII, 28029 Madrid, Spain; 6Departamento de Pediatría, Radiología y Medicina Física, Universidad de Zaragoza, 12, 50009 Zaragoza, Spain

**Keywords:** rapid weight gain, infant feeding, breastfeeding, body mass index z-score

## Abstract

We aimed to study growth patterns according to rapid weight gain (RWG) and infant feeding practices during the first 120 days and whether infant feeding practices mediated the association between RWG in the first semester of life and subsequent body mass index (BMI) z-score in children from age 1 to 6. (1) Methods: 862 children from the Growth and Feeding during Lactation and Early Childhood in Children of Aragon study (CALINA in Spanish) were examined. Repeated-measures ANOVA analyses were conducted to assess growth trajectories according to RWG and type of feeding practice. The product of coefficients mediation method was used to assess the potential contribution of infant feeding practices to the association between RWG and BMI z-score. Mediation models were conducted using IBM SPSS-PROCESS Statistics for Windows, Version 26.0. Armonk, NY: IBM Corp. (2) Results: BMI and weight z-score trajectories were significantly higher in the RWG group and the formula-fed group. No significant differences were found regarding height. Infant feeding practices did not mediate the association between RWG and BMI z-score but were associated with BMI at 6 years. (3) Conclusions: Infant feeding practices and RWG determine different growth trajectories of BMI and weight during childhood. Although infant feeding practices did not mediate the association between early RWG and BMI later in life, formula feeding is independently related to higher BMI growth patterns later in childhood.

## 1. Introduction

Obesity prevalence has increased by 100% in the last 40 years worldwide and it does not only affect adults, but also children; in fact, over 40 million children under 5 are already overweight and are therefore predisposed to later overweight or obesity during adulthood [1]. According to the World Health Organization (WHO) European Childhood Obesity Surveillance Initiative (COSI) study, overweight and obesity prevalence in European children from 6 to 9 years old oscillates among countries between 19.3–49.0% of boys and 18.4–42.5% of girls when following the 2007 WHO growth references, with Belgium, the Czech Republic, and Norway being the countries with the lowest rates of overweight and obesity and Portugal, Slovenia, and Italy being the countries with the highest rates [2].

Obesity, defined by the WHO as an abnormal or excessive fat accumulation presenting a risk to health [3], has a multifactorial origin, involving mainly genetic and environmental factors [4]. Early-life environmental factors during the perinatal period are recognized as major factors in shaping obesity risk, both in childhood and even later in adulthood [5,6]. In particular, parental exposures such as higher pre-pregnancy body mass index (BMI), prenatal tobacco exposure, excessive gestational weight gain, high birth weight, and infant rapid weight gain (RWG) are factors strongly associated with future childhood obesity [5,7].

From the early stages of life, nutrition plays an important role in growth and development. Breastfeeding is considered by pediatric organizations, such as the Italian [8] and the American [9] Associations of Pediatrics and the WHO [10], as the best method of nourishment for babies and infants during their first year of life. However, there is still controversy about whether breastfeeding can be considered as a protective factor against later obesity [11], as breastfeeding is probably confounded by the effect of socioeconomic status of families [12]. Furthermore, rapid or excessive weight gain during the first months of life seems to be an important risk factor for developing childhood overweight or obesity [5]. In this sense, type of feeding is of relevance, given that formula feeding has been identified as a risk factor for RWG during the first 6 months of life [13], and, on the other hand, breastfeeding has been associated with lower odds of rapid increase in weight when evaluated in different time periods during the first 24 months [14,15]. In addition to isolated differences, there have been differences in growth trajectories in terms of weight and BMI z-scores between breastfed and formula-fed infants, with breastfed infants having lower weight-for-age z-score (WAz) and BMI-for-age z-score (BAz) trajectories [16].

These two factors, exclusive breastfeeding or formula feeding and RWG, occur in parallel and therefore might be inter-related. An example of evidence of this is the fact that formula-fed infants, as opposed to breastfed infants, are more likely to experience RWG [17]. However, there is little information about the impact of the type of infant feeding on the association between RWG and further risk of obesity development during childhood.

In current literature, most studies have shown associations between feeding practices and RWG. Meanwhile, other studies have focused on the effects that RWG has on growth trajectories and later overweight risk. Therefore, in order to try and explain whether type of feeding mediates the established association between RWG and body weight and BMI, the aim of this study is to describe growth patterns according to RWG (whether they were classified as RWgainers or not from 0 to 6 months of life) and type of infant feeding during the first 120 days (exclusive breastfeeding, formula feeding, or mixed feeding) and to assess the mediation effect of the type of feeding during the first 4 months of life on the association between RWG and BMI z-score from 1 to 6 years of age.

## 2. Materials and Methods

### 2.1. Design and Study Population

The study Growth and Feeding during Lactation and Early Childhood in Children of Aragon (Crecimiento y ALimentación en la Infancia en Niños Aragoneses, CALINA in Spanish) is an ongoing birth cohort study whose sampling design is described elsewhere in detail [18]. Briefly, CALINA is a representative birth cohort study of children born between March 2009 and February 2010 in the Autonomous Region of Aragon in Spain. Participants were recruited from primary care centers, where trained pediatricians oversaw measurements. The main objective of the CALINA study was to assess growth patterns, body composition, and feeding aspects in this population and to examine prenatal, postnatal, and socioeconomic factors that could have an influence on them.

In total, 1630 families were contacted during the first 2 weeks of life of the newborns in their primary care centers and invited to participate in the CALINA study, of which 1602 accepted to participate (acceptance rate: 98%). Exclusion criteria included presenting any malformations, diseases, or physical disabilities. After families accepted to participate in the study, data regarding prenatal factors and birth characteristics were obtained from mothers and their newborn children clinical histories, and direct interviews with the families. Perinatal information from children was obtained after enrollment, and children were periodically re-examined in primary care centers at 2 weeks, monthly (at 1, 2, 4, 6, and 9 months), and yearly (at 1, 2, 3, 4, 5, and 6 years of age). In every assessment, besides clinical evaluation of health indicators, growth indicators (weight and length or height) were measured by pediatricians and nurses that had previously been trained for consistency between measurements. Furthermore, information regarding feeding practices was obtained in order to know what type of feeding children were receiving each month.

For this study, preterm infants (with 36 weeks of gestational age or less at birth) and those without complete data regarding infant feeding practice, weight or height at 6 years (72 months) were excluded from the analysis (*n* = 740). In the end, the present study included 862 children. Differences between socioeconomic characteristics of the included and the excluded sample were analyzed, and significant differences were observed in maternal age (32.3 ± 5.01 years in the included vs. 31.21 ± 5.3 years in the excluded sample, *p* = 0.000), origin (14.6% of immigrants in the included sample vs. 33.2% in the excluded sample, *p* = 0.000), and education (25.2% of low-educated mothers in the included sample vs. 32.3% in the excluded sample and 39.8% of high-educated mothers vs. 32.8% in the excluded sample, *p* = 0.002).

The study was performed following the ethical guidelines of the Declaration of Helsinki 1964 [19]. Parents or legal guardians provided signed consent for participation. Ethical approval was obtained in June 2008 (P108/0021) from Aragon’s Committee of Ethics in Clinical Research (Comité de Ética de la Investigación de la Comunidad de Aragón, CEICA).

### 2.2. Measurements

#### 2.2.1. Anthropometry (Outcome)

The primary outcomes were z-scores for BMI, weight and length/height assessed at 1, 2, 3, 4, 5, and 6 years old. Body weight was measured to the nearest 10 g using a weight scale/bioelectric impedance instrument (Tanita ^®^ BC-418, Corporation of America, Inc., IL, USA). Child length up to 2 years was measured using a recumbent board. For children older than 2 years, barefoot body height was measured to the nearest 0.1 cm standing up in a fasting state and wearing light clothes using a stadiometer (SECA 225, SECA, Hamburg, Germany). Both height and weight were obtained by trained staff. BMI-, weight- and height-for-age-and-sex z-scores (BAz, WAz, and HAz, respectively) were calculated with the Anthro Software v 3.2.2, Geneva, Switzerland and Anthro Plus Software v 1.0.4, Geneva, Switzerland, according to the WHO growth references [20].

#### 2.2.2. Rapid Weight Gain (Exposure)

The primary exposure was RWG, a dichotomous variable defined as “no” or “yes”. RWG is defined as a positive change in the WAz greater than 0.67 between two different ages in childhood [21]. In this study, we determined RWG in the first semester of life, from birth to 6 months of age. Determination of z-score values of weight for age at birth and at 6 months for girls and for boys was performed using the WHO Anthro Software ^®^, according to the WHO growth standards of 2006–2007 [20].

#### 2.2.3. Type of Infant Feeding (Mediator)

Children were classified according to the type of infant feeding they received during the first 120 days of life into 3 categories: 1 = Formula, 2 = Mixed feeding, and 3 = Breast milk. Formula feeding included children that received only formula during the first 120 days of life, and the third category included children that had exclusively received breast milk during this time. The second category, mixed feeding, included children that received both formula and breast milk. This group included those that received 1 or 2 months of exclusive breastfeeding and then a combination of both or those that received breast milk and formula since they were born.

#### 2.2.4. Covariates

Covariates were identified a priori as potential confounders based on known associations between RWG and further weight gain. These covariates included birth weight, gestational age, maternal and paternal education, maternal and paternal BMI before pregnancy, parental origin, and maternal smoking during pregnancy. (I) Birth weight in grams was a continuous variable. (II) Gestational age in weeks was a continuous variable. (III) Maternal and paternal education level: parents reported their highest level of studies achieved (none, basic, graduate, or postgraduate), which were re-categorized into 3 categories: low (none or basic), medium (graduate), or high (postgraduate). (IV) BMI from parents before pregnancy was obtained by a face-to-face interview in which they reported weight and height and was thereafter calculated as weight in kilograms divided by the square of height in meters (kg/m^2^). (V) Parental ethnicity: families were classified into 2 categories according to the origin of their parents, which could be Spanish or immigrant, and (VI) Maternal smoking during pregnancy: this variable was a dichotomous variable that indicated whether the mother had smoked during pregnancy, regardless of the number of cigarettes.

### 2.3. Statistical Analyses

Statistical analyses were carried out using IBM SPSS Statistics (IBM SPSS Statistics for Windows, Version 26.0. Armonk, NY: IBM Corp.). A descriptive analysis was conducted using mean and standard deviation for continuous variables and frequencies and percentages for categorical variables according to having developed RWG in the first semester of life.

To test whether the trajectories of BAz, WAz, and HAz differed based on the presence of RWG and the type of infant feeding, we used repeated-measures ANOVA in two separate models [22]. The first model included RWG as the main effect and BAz, WAz, and HAz at birth, 6 months, 1, 2, 3, 4, 5, and 6 years as the repeated-measures variable and the second model included infant feeding practice as the main effect and birth, 6-month, and yearly BAz, WAz, and HAz as the repeated-measures variable. These models were also conducted adjusting by birth weight, gestational age at birth, maternal education, maternal and paternal BMI before pregnancy, parental origin, and maternal smoking during pregnancy.

### 2.4. Mediation Analyses

To assess whether the associations between RWG and BMI z-score were mediated by infant feeding, the following steps were performed.

To qualify as a mediator, the presumed mediator (M) has to be associated with the predictor variable (a-pathway) and also with the outcome variable (b-pathway) [23]. This was assessed by performing bivariate correlations. Then, based on the hypothesis that infant feeding practice could mediate the associations between RWG and BAz during early childhood, the mediation models were examined using the PROCESS macro 3.1.4 software for SPSS [24]. In PROCESS, Model 4 software was applied for simple mediations. The independent variable was RWG, the dependent variable was BAz at 6 years, and the potential mediator variable was infant feeding practice. The covariates included were birth weight, gestational age at birth, maternal education, maternal and paternal BMI at child’s birth, parental origin, and maternal smoking during pregnancy. Indirect effects were computed using a bias-corrected bootstrapping procedure (5000); if the 95% confidence interval did not include 0, it meant that the mediation effect was significant.

## 3. Results

Characteristics of rapid weight gainers and non-rapid weight gainers included in the analysis are shown in Table 1. Mean of birth weight was significantly higher in the non-rapid weight gainers group (*p* < 0.001), but from 6 months onward, children in the RWG group presented significantly higher BAz at 6, 12, 24, 36, 60, and 72 months (*p* < 0.01 in all cases). Differences in BAz at 48 months were not significantly different across groups.

Trajectories of BAz, WAz, and HAz in rapid weight gainers and non-rapid weight gainers and according to infant feeding practice are shown in Figure 1 and Figure 2, respectively.

Figure 1 shows the non-adjusted repeated-measures ANOVA models of z-scores of BMI, weight, and length-height. *p*-values from Box’s tests are presented on the z-score trajectory of each figure. This value represents the significance of rejecting the assumption of equality variance–covariance which is significant and therefore indicates that there are differences across trajectories of groups. Models were also performed adjusting for covariates that we identified a priori as potential confounders based on known associations with feeding type and BAz.

Participants in the RWG group presented significantly higher BAz and WAz from birth to 72 months (*p* = 0.006 and *p* = 0.043, respectively). HAz trajectories did not differ across groups (*p* = 0.828). After adjusting for covariates (birth weight, gestational age at birth, maternal and paternal education, maternal and paternal BMI at child’s birth, ethnicity, and maternal smoking during pregnancy), BAz trajectories still differed between groups (*p* = 0.001). WAz and HAz trajectories were not statistically different between groups (*p* = 0.16 and *p* = 0.89, respectively).

Trajectories of growth indicators according to infant feeding practice in the first 120 days of life (Figure 2) showed that formula-fed infants had higher BAz and WAz compared to infants fed with mixed-feeding or breast milk (*p* < 0.001 and *p* = 0.005, respectively). HAz trajectories did not differ across categories (*p* = 0.560). After adjusting for birth weight, gestational age, maternal and paternal education, maternal and paternal BMI, parental origin, and maternal smoking, the trajectories of BAz and WAz remained statistically different among infant feeding practice groups (*p* < 0.01 in both cases). HAz trajectory was not statistically different between groups (*p* > 0.05) in the adjusted model.

### Mediation Results

Given that the presumed mediator (M) has to be associated with the predictor variable (a-pathway) and also with the outcome variable (b-pathway) to qualify as a mediator; Pearson correlations were performed to confirm this by assessing if there are associations between infant feeding practices and RWG development in the first semester of life, infant feeding practices, and BMI z-score at 1, 2, 3, 4, 5, and 6 years (Table 2). In this step, we obtained that RWG was significantly correlated with BAz at 1, 2, 3, 5, and 6 years and that infant feeding practices (1 = Breast milk, 2 = Mixed feeding, and 3 = Formula milk) were positively associated with RWG development, which indicated that the mediation analyses could only be performed considering BAz at 6 years as the dependent variable.

Results of mediation can be found in Table 3 and a graphical explanation of the model is presented in Figure 3. Non-adjusted mediation analyses showed that Pathway A’ (association between RWG and infant feeding practice), indicating that formula intake was positively associated with RWG development. Pathway B’ (association between type of feeding and BAz at 6 years) was also positive and significant. Pathway C, which represents the association between RWG and BAz was positive and significant (*p* < 0.02), as was Pathway C’, which represents the previous associations but considering the mediator as a covariate. After adjustments, the significance in the pathways was the same as in the unadjusted model, except for Pathway A’, which was not significant.

The indirect effect of the potential mediator, type of feeding, indicated that the mediating effect of infant feeding practice on the relationship between RWG and BMI z-score (AB coefficient) was not statistically significant in either the unadjusted or the adjusted model, which means that there is no significant difference in the association between RWG and BAz at 6 years, depending on the type of feeding children received in the first 120 days of life. In general, there is a statistically significant direct effect between RWG and BAz from 0 to 6 years old, regardless of covariates, but infant feeding practice during the first 120 days did not mediate this association.

## 4. Discussion

The present study aimed to describe growth patterns according to RWG (whether they were classified as RWgainers or not from 0 to 6 months of life) and type of infant feeding during the first 120 days (exclusive breastfeeding, formula feeding, or mixed feeding) and to study whether these feeding practices mediate the association between RWG in the first semester of life and later BAz in children from 1 to 6 years. We found that BAz and WAz trajectories were significantly higher in rapid weight gainers compared to non-rapid weight gainers. Regarding infant feeding practices, BAz and WAz trajectories were significantly higher in the formula-fed group than breastfed or mixed-fed children. However, no significant differences were found for height in either of the analyses. Infant feeding practices for the 120 first days of life did not mediate the association between RWG and BAz at 6 years, although it appeared to be independently associated with BAz at this age.

An important aspect of the present study is that z-scores have been used to assess growth in children, which is widely recognized as the best system for analysis and presentation of anthropometric data because of its advantages compared to other methods. Furthermore, given that they are sex-independent, their use permits the evaluation of children’s growth status by combining sex and age groups [25] and has been widely used in recent studies with similar aims [26,27,28].

In the present investigation, and after adjusting for birth weight, gestational age at birth, parental education, parental BMI before pregnancy, ethnicity, and maternal smoking during pregnancy, the type of infant feeding (breastfeeding, mixed feeding, or formula feeding) during the first 120 days did not have a direct effect on RWG. However, an important direct effect of breastfeeding on long term BAz (BMI at 6 years old) was visible. Even though this study did not find the causes behind this association, results showed a small but significant protective effect of breastfeeding on the development of excess adiposity (BAz). Additionally, this effect appeared to be independent of the effect that breastfeeding had on RWG.

According to some studies, formula-fed infants are more likely to have RWG than breastfed infants. However, in a recent meta-analysis, only limited evidence (if any) was found in the association between formula feeding practices and RWG in infancy [29]. In our analysis, the association between type of infant feeding and RWG from 0 to 6 months of age showed a statistically significant effect only in the non-adjusted models. In fact, this association was no longer significant after adjusting for the abovementioned covariates. The increased risk of RWG in formula-fed infants may be due to a number of different factors, such as feeding to schedule versus feeding on demand [17]. It may also be due to the mode of milk delivery [30], since breastfeeding may be associated with better appetite regulation in the long term in comparison to bottle-feeding, regardless of the type of milk in the bottle [31]. It can be suggested that formula feeding is not directly associated with RWG, but some behaviors related to it, such as the use of high protein formula or adding cereals into the bottle, may be related [29].

Regarding the relationship between infant feeding practices and BAz, there was only a negative significant association between breastfeeding and BAz at 6 years old in the fully adjusted model. In other studies, breastfeeding was only significant in raw models, but when adjusting for important confounders, this association did not remain significant [32]. In our study, although this protective effect of breastfeeding could only be seen after several years, the effect was small but significant. Similarly, delaying the introduction of bottle feeding had a protective effect against obesity at 6 years of age in a sample of Spanish children, highlighting the need for greater support of breastfeeding to avoid future childhood obesity [11,33].

Similar to other studies [34,35,36], we found an important direct effect of RWG on children’s BMI and, moreover, not mediated by breastfeeding during the first 120 days. This effect persisted over the years and remained significant at 6 years old. Even though the mechanism through which infant RWG programs subsequent adiposity remains unclear, it has been speculated that this association may be influenced by maternal factors and birth weight.

In this study, the average birth weight was significantly lower among rapid weight gainers compared to non-rapid weight gainers. Because of the potential influence of low birth weight on accelerated postnatal catch-up growth [37], which occurs typically in the first 24 months of postnatal life in infants with low birth weight, birth weight was included as a covariate in all models.

Nevertheless, from 6 months onwards, those rapid weight gainers presented a higher BAz at all ages (which remained significant after adjustments) and WAz (which did not remain significant after adjustments). Particularly, it has been shown that infants of low birth weight are more likely to have higher adrenal androgen levels, insulin resistance, and central fat deposition. They are therefore prone to weight gain [38], which may explain at least part of our results. In this line, it is important to note that our sample also differed in terms of birth weight, and even though it was included in the models as a covariate, it may be another starting point that determines different growth trajectories between groups.

We also adjusted for different maternal factors and birth weight and found a statistically significant result between RWG and BMI, suggesting that there must be other factors that are playing a role independently. Beyond birth weight, gestational age at birth, parental education, parental BMI before pregnancy, ethnicity, and maternal smoking during pregnancy, factors that may play an important role include dietary intake [39], eating behaviors [40], parental feeding practices [41], and physical activity levels [42].

Finally, even though there is a significant difference in growth trajectories, especially in the BAz, according to RWG in the first semester of life and infant feeding practices, no mediation effect of infant feeding practice was observed in this association.

### Strengths and Limitations

To the best of our knowledge, this is the first study that has analyzed the mediation effect that infant feeding practices might have in the association with BAz during childhood in a large cohort of Spanish children followed from birth to 6 years old, taking into account perinatal factors from their parents. The prospective collection of data on a wide range of risk factors extending from pregnancy through infancy and the ability to adjust for several important confounding socioeconomic factors are among the strengths of this study.

However, there are some limitations to this research that should be acknowledged. Firstly, the included children are not representative of the Spanish population since the region in which the study took place (Aragon) covers a limited geographic area within the country (northeast) and results might not be extrapolated to the whole Spanish population. Secondly, an analysis of further confounders highly associated with obesity levels, such as dietary patterns, sedentary behaviors, physical activity, sleep duration, or family income, were not included in this study. Thirdly, there are some measures that were reported by parents that could imply a reliance problem (such as parental weight and height and their education). Furthermore, information regarding specific feeding practices, such as feeding to schedule or feeding on demand and the addition of cereals to bottles, was not included in the analyses, and it may be important to consider their effect on growth and weight gain. Moreover, a selection bias cannot be precluded as there were participants (mainly children whose parents were originally from Eastern European countries, Africa, and Latin America and had lower parental education) who did not complete all information required or did not continue the study at follow-up. Furthermore, feeding practice report may be prone to bias, given that self-administered questionnaires and health staff registrations on a monthly or less frequent basis have shown to be measurements that may overestimate the duration of exclusive breastfeeding [43]. Future studies should consider addressing other potential factors, such as dietary intake, physical activity, and sleep, which are lifestyle factors that may also affect growth trajectories in addition to type of feeding during the first months of life. 

## 5. Conclusions

Both infant feeding practices and RWG determine different growth trajectories in terms of BMI and weight during childhood. Even though infant feeding practices do not seem to mediate the association between early RWG and BMI later in life, formula feeding is independently related to higher BMI growth patterns later in childhood.

## Figures and Tables

**Figure 1 nutrients-12-03178-f001:**
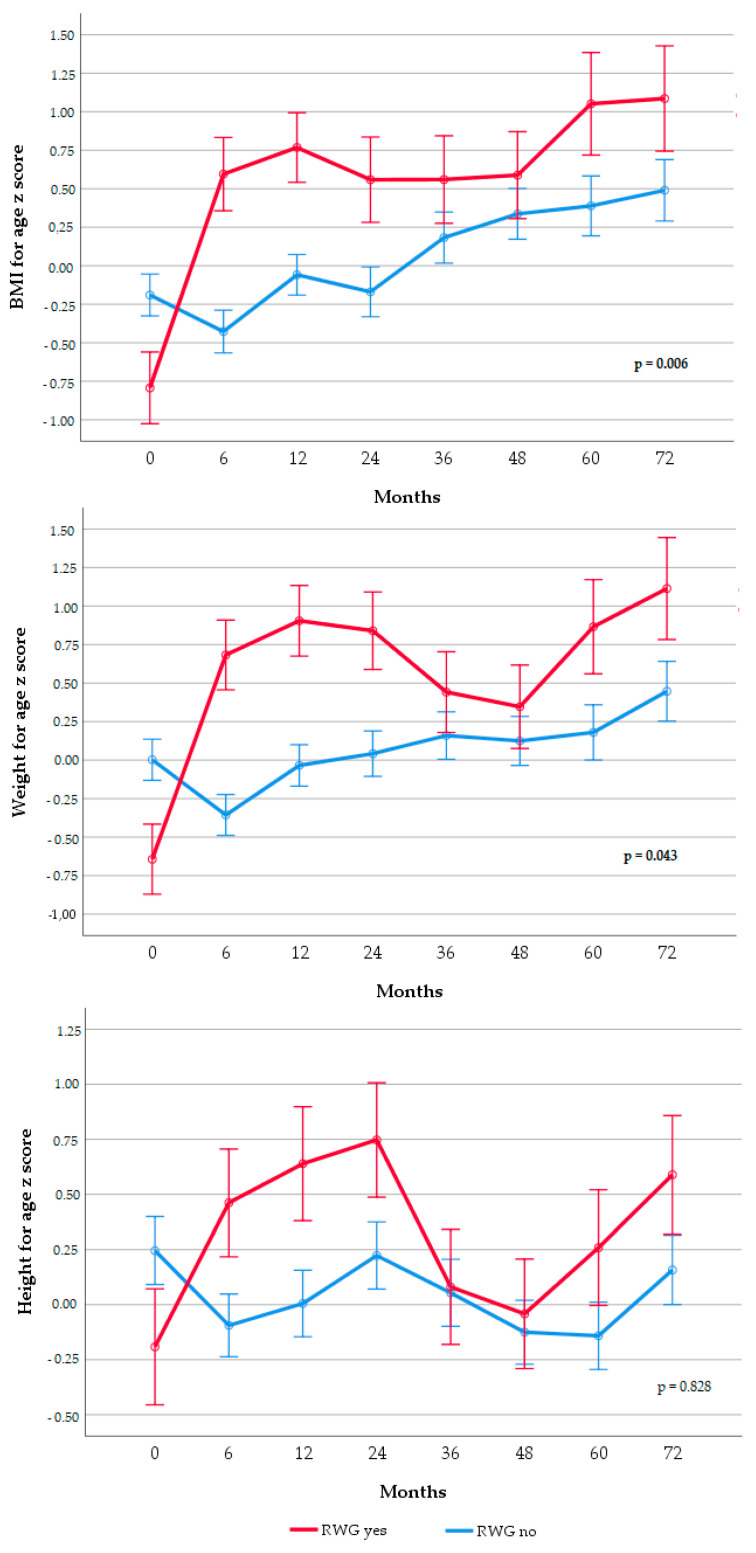
Trajectories of z-scores for infant BMI, weight, and height according to RWG from 0 to 6 months in infants. Results are displayed as estimated mean z-scores SEMs from repeated-measures ANOVA. BMI, body mass index; RWG, rapid weight gain. After adjusting by gestational age at birth, birth weight, maternal education, ethnicity, paternal and maternal BMI, and smoking during pregnancy, significant differences remained in BAz (*p* = 0.01), but not in WAz (*p* = 0.16) or HAz (*p* = 0.89). BMI: body mass index, RWG: rapid weight gain.

**Figure 2 nutrients-12-03178-f002:**
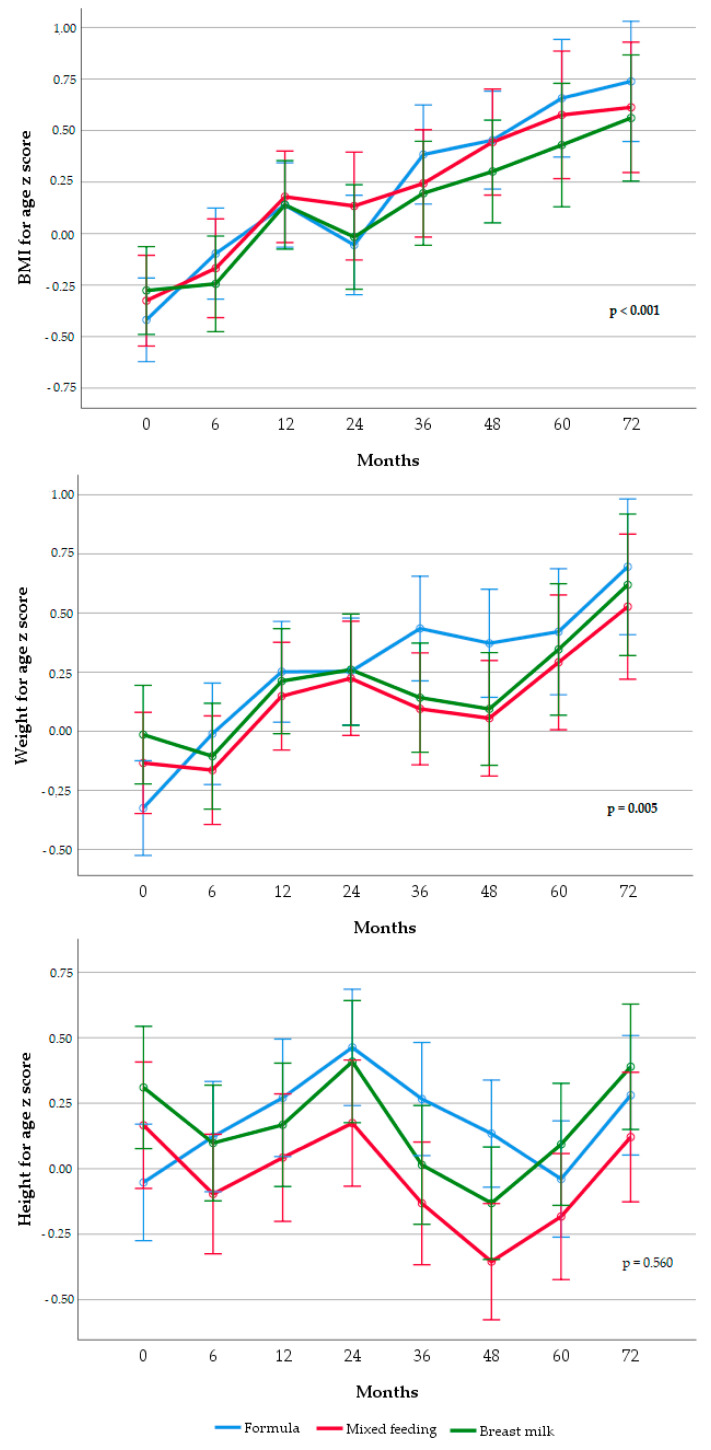
Trajectories of z-scores of infant BMI, weight, and height according to infant feeding practice in the first 120 days of life. Results are displayed as unadjusted estimated mean z-scores +/− 2SE from repeated-measures ANOVA. After adjusting by gestational age at birth, birth weight, maternal education, ethnicity, paternal and maternal BMI, and smoking during pregnancy, significant differences remained in BAz (*p* < 0.01) and WAz (*p* < 0.01) but not in HAz (*p* = 0.33). BMI, body mass index; RWG, rapid weight gain, SE, standard error.

**Figure 3 nutrients-12-03178-f003:**
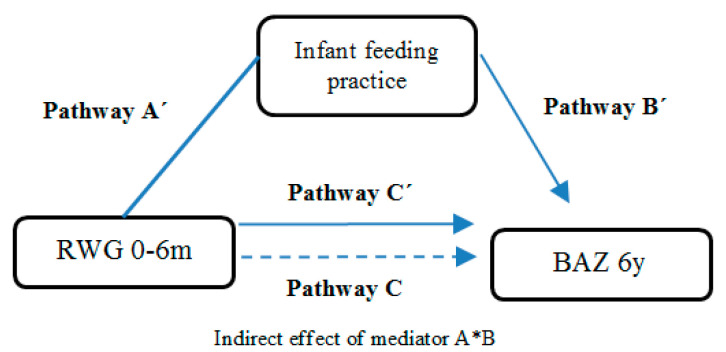
Graphical illustration of the significant interactions between rapid weight gain, type of feeding, and BAz at 6 years.

**Table 1 nutrients-12-03178-t001:** Descriptive characteristics of participants according to rapid weight gain.

	Total	RWG Yes	RWG No	*p*-Value
*%* (*n*)	862	28.31 (244)	71.69 (618)	
Girls	47.2 (407)	48.4 (118)	46.8 (289)	0.672
Boys	52.8 (455)	51.6(126)	53.2 (329)	
Gestational age at birth mean ± SD	39.16 ± 1.33	38.53 ± 1.46 (244)	39.41 ± 1.91 (618)	**0.000** **0.000**
Small for gestational age, % (*n*)	11.4 (98)	26.6 (65)	5.3 (33)	
Normal for gestational age, % (*n*)	88.6 (764)	66.3 (179)	94.7 (585)	
Birth weight mean ± SD (*n*)	3263.41 ± 0.98 (862)	2973.12 ± 406.98 (244)	3378.02 ± 415.11 (618)	**0.000**
BAz mean ± SEM (*n*)				
At birth	–0.31 ± 0.03 (855)	−0.87 ± 0.06 (242)	−0.09 ± 0.04 (613)	**0.000**
6 months	−0.08 ± 0.04 (851)	0.61 ± 0.07 (239)	−0.35 ± 0.04 (612)	**0.000**
12 months	0.25 ± 0.03 (840)	0.75 ± 0.06 (237)	0.05 ± 0.04 (603)	**0.000**
24 months	0.11 ± 0.04 (802)	0.60 ± 0.07 (220)	−0.07 ± 0.04 (582)	**0.000**
36 months	0.26 ± 0.04 (771)	0.43 ± 0.08 (214)	0.20 ± 0.05 (557)	**0.013**
48 months	0.41 ± 0.04 (753)	0.54 ± 0.09 (213)	0.36 ± 0.05 (540)	0.064
60 months	0.62 ± 0.07 (316)	1.07 ± 0.14 (88)	0.44 ± 0.08 (228)	**0.000**
72 months	0.62 ± 0.04 (862)	1.03 ± 0.08 (244)	0.45 ± 0.05(618)	**0.000**
Delta in WAz mean ± SD (*n*)	0.11 ± 1.05 (862)	1.34 ± 0.58 (244)	−0.38 ± 0.76 (618)	**0.000**
Infant feeding practice in the 1st 120 days, % (*n*)				0.136
Formula	34.2 (295)	38.5 (94)	32.5 (201)	
Mixed (Formula + Breast milk)	27.7 (239)	28.3 (69)	27.5 (170)	
Breast milk	38.1 (328)	33.2 (81)	40.0 (247)	
Maternal education, % (*n*)				**0.009**
None or basic	24.6 (212)	32.4 (77)	22.3 (135)	
Intermediate	34.6 (296)	30.7 (73)	36.9 (223)	
High	38.9 (335)	37.0 (88)	40.8 (247)	
Paternal education, % (*n*)				0.326
None or basic	29.8 (257)	34.5 (79)	29.6 (178)	
Intermediate	42.0 (362)	42.8 (98)	43.9 (264)	
High	24.5 (211)	22.7 (52)	26.5 (159)	
Parental origin/ethnicity, % (*n*)				0.325
Spanish	84.5 (728)	82.9 (194)	85.6 (518)	
Immigrant	15.3 (132)	17.1 (40)	14.4 (87)	
Parental BMI, kg/m^2^				
Mothers mean ± SD (*n*)	23.65 ± 4.45 (836)	24.03 ± 4.76 (239)	23.49 ± 4.32 (597)	0.117
Fathers mean ± SD (*n*)	26.19 ± 3.38 (805)	26.37 ± 3.26 (223)	26.11 ± 3.43 (582)	0.332
Maternal smoking during pregnancy, % (*n*)				0.094
Yes	18.2 (157)	21.7 (53)	16.8 (104)	
No	81.2 (705)	78.3 (191)	83.2 (514)	
Center, % (*n*)				0.226
Huesca	73.8 (636)	77.9 (190)	72.2 (446)	
Teruel	17.2 (148)	14.8 (36)	18.1 (112)	
Zaragoza	9.0 (78)	7.4 (18)	9.7 (60)	

Abbreviations: BAz, body mass index for age z-score, BMI, body mass index, RWG, rapid weight gain, SEM, standard error of measurements. RWG was defined as a positive change (>0.67) in the weight-for-age z-score in the established periods. Statistical significance was set at *p* < 0.05, significant values are shown in bold. Categorical variables were assessed with Pearson’s chi-square tests. Means were compared with *t*-test.

**Table 2 nutrients-12-03178-t002:** Bivariate correlations among rapid weight gain, infant feeding practices, and BAz in children from the CALINA ^a^ study.

BMI *z*-Scores ^b^	RWG ^c^	Maternal Education ^d^	Paternal Education ^d^	Birth Weight	Maternal BMI ^e^	Paternal BMI ^e^	Infant Feeding Practice ^f^
BAz 1 years	0.304 ***	−0.061 *	−0.046	0.207 ***	0.131 ***	0.112 ***	−0.007
BAz 2 years	0.256 ***	−0.062 *	−0.055 *	0.215 ***	0.151 ***	0.117 ***	−0.006
BAz 3 years	0.077 **	−0.063 *	−0.52 *	0.020	0.002	−0.006	0.019
BAz 4 years	0.047	−0.038	−0.040	0.060	−0.008	0.021	0.023
BAz 5 years	0.154 **	−0.041	−0.036	0.189	0.227 ***	0.189 ***	0.057
BAz 6 years	0.193 ***	−0.117 ***	−0.122 ***	0.141 ***	0.208 ***	0.192 ***	0.093 **
Infant feeding practice ^f^	0.084 **				-	-	

^a^ CALINA, Growth and Feeding during Lactation and Early Childhood in Children of Aragon (Crecimiento y ALimentación en la Infancia en Niños Aragoneses in Spanish), ^b^ BMI z-score (BAz), body mass index z-score for age and sex according to World Health Organization (WHO) growth standards; RWG, rapid weight gain; BMI, body mass index; y, years. ^c^ RWG, Rapid weight gain was defined as a positive change of >0.67 in the BMI z-score ^a^. ^d^ Maternal and paternal education were classified as none or basic, medium, or high^. e^ Maternal and paternal BMI (body mass index) before pregnancy was calculated from self-reported weight and height with the formula kg/m^2^. ^f^ Infant feeding practice was a categorical variable classified as 1 = Breast Milk, 2 = Mixed feeding (formula milk and breast milk), and 3 = Formula milk, indicating the type of practice received during the first 120 days. Note: Bivariate Pearson correlations are presented. * Correlation is significant at the 0.05 level, ** Correlation is significant at the 0.01 level, *** Correlation is significant at the 0.001 level.

**Table 3 nutrients-12-03178-t003:** The mediation of infant feeding practice during the first 120 days of life between rapid weight gain during the first semester of life and BAz at 6 years.

	Pathway A’ β (95% CI)	Pathway B’ β (95% CI)	Pathway C’ β (95% CI)	Pathway C β (95% CI)	Indirect Effect of Mediator A*B β, SE (95% CI)	Mediation
^(A)^*n* = 862	**0.128 (0.002; 0.254) ***	**0.118 (0.021; 0.215) ***	**0.562 (0.379; 0.745) *****	**0.577 (0.394; 0.760) *****	0.015, SE = 0.01 (−0.001; 0.039)	-
^(B)^*n* = 767	−0.036 (−0.185; 0.112)	**0.114 (0.015; 0.212) ***	**0.784 (0.579; 0.990) *****	**0.780 (0.575; 0.986) *****	−0.004. SE = 0.0094 (−0.026; 0.013)	-

^(A)^ Unadjusted simple mediation analysis, ^(B)^ Simple mediation analysis adjusted by birth weight, gestational age at birth, maternal education, maternal and paternal BMI at child’s birth, ethnicity, and maternal smoking during pregnancy. Pathway A’: Association between RWG and infant feeding practice. Pathway B’: Association between infant feeding practice and BAz at 6 years. Pathway C’: Direct association of RWG and BAz after adjustment for mediator (infant feeding practice). Pathway C: The total effect (c) shows the association between RWG in the 1st semester of life and BAz at 6 years. A*B: Indirect effect of the infant feeding practice on the association between RWG and BAz. Rapid weight gain was defined as a positive change of >0.67 in the z-score of BMI for age and sex according to World Health Organization (WHO) growth standards. Infant feeding practice was a categorical variable classified as 1 = Breast milk, 2 = Mixed feeding (formula milk and breast milk), and 3 = Formula milk. BMI z-score was calculated according to the WHO growth standards. Abbreviations: RWG, rapid weight gain; BAz, body mass index z-score.; CI: confidence interval. Statistically significant results are shown in bold font. Note. Unstandardized coefficients are presented, SE; standard error, CI; confidence interval, * *p*-value is significant at the 0.05 level and *** *p*-value is significant at the 0.001 level, - = Not significant indirect effects.

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
