# Peer review of "Rapid Weight Gain, Infant Feeding Practices, and Subsequent Body Mass Index Trajectories: The CALINA Study"

_nutrients, 2020, doi:10.3390/nu12103178_

Round 1

Reviewer 1 Report

Thank you for your work. Overall, I felt that this was a well-written article that was easy to understand. I have just a few notes.

Lines 41-42 may need some grammatical edits. 

Intro (or maybe Discussion): Readers may benefit from an explanation about the different aspects (and previous use in studies) of using BMI, weight, or height -for age z-scores along the 1-6 year age range. 

Methods (rapid weight gain): Consider including a brief explanation about why weight-for-age was chosen as the metric. This may fit better in the introduction.

Results: Is there data about they type of formula and ratio of formula:breast milk in the mixed group? 

Discussion (line 282-285): Consider expanding the explanation about some of the mechanisms, and clarify which mechanism may explain the study's outcome.  Also, reconsider the last sentence's inclusion of higher protein content of formulas as a behavior. Is there data about other feeding practices used by the formula group in the current study?

Discussion (starting line 305): Consider explaining catch-up growth, a compensatory mechanism, in low-birth infants and subsequent metabolic programming.

Author Response

Response to Reviewer 1 Comments

Point 1: Lines 41-42 may need some grammatical edits. 

Response 1: Changes have been performed; we hope now the idea is clearer.

Point 2: Intro (or maybe Discussion): Readers may benefit from an explanation about the different aspects (and previous use in studies) of using BMI, weight, or height -for age z-scores along the 1-6-year age range. 

Response 2: This is an important idea; we have included this in the discussion section in lines 331-335:

“An important aspect of the present study is that z-scores have been used to assess growth in children, which is widely recognized as the best system for analysis and presentation of anthropometric data because of its advantages compared to the other methods and given that they are sex-independent, their use permits the evaluation of children´s growth status by combining sex and age groups [25] and has been widely used in recent studies with similar aims [26-28]”

References:

  1. World Health Organization. Global Database on Child Growth and Malnutrition. The Z-score or standard deviation classification. Availabe online: https://www.who.int/nutgrowthdb/about/introduction/en/index4.html (accessed on 29/09/2020).
  2. Woo, J.; Sucharew, H.; Su, W.; Khoury, P.; Daniels, S.; Kalkwarf, H. Infant Weight and Length Growth Trajectories Modeled Using Superimposition by Translation and Rotation Are Differentially Associated with Body Composition Components at 3 and 7 Years of Age. The Journal of pediatrics 2018, 196, doi:10.1016/j.jpeds.2017.12.077.
  3. Zheng, M.; Bowe, S.; Hesketh, K.; Bolton, K.; Laws, R.; Kremer, P.; Ong, K.; Lioret, S.; Denney-Wilson, E.; Campbell, K. Relative effects of postnatal rapid growth and maternal factors on early childhood growth trajectories. Paediatric and perinatal epidemiology 2019, 33, doi:10.1111/ppe.12541.
  4. Salahuddin, M.; Pérez, A.; Ranjit, N.; Hoelscher, D.; Kelder, S. The associations of large-for-gestational-age and infant feeding practices with children's body mass index z-score trajectories: the Early Childhood Longitudinal Study, Birth Cohort. Clinical obesity 2017, 7, doi:10.1111/cob.12201.

Point 3: Methods (rapid weight gain): Consider including a brief explanation about why weight-for-age was chosen as the metric. This may fit better in the introduction.

Response 3: Thanks for this comment. References 13 – 15 have been included as examples of the use of weight-for-age as a growth indicator.

“In this sense, type of feeding is of relevance, given that formula feeding has been identified as a risk factor for RWG during the 6 months of life [13] and, in counterpart, breastfeeding has been associated with lower odds of rapid increase in weight when evaluated in different time periods during the first 24 months [14,15]. Moreover, not only isolated differences have been observed, but also differences in growth trajectories in terms of weight and BMI z scores between breastfed and formula-fed infants, being those that had been breastfed, the ones with lower WAZ and BAZ trajectories [16].”

References:

  1. Yang, S.; Mei, H.; Mei, H.; Yang, Y.; Li, N.; Tan, Y.; Zhang, Y.; Zhang, D.; Zhang, Y.; Peng, A., et al. Risks of maternal prepregnancy overweight/obesity, excessive gestational weight gain, and bottle-feeding in infancy rapid weight gain: evidence from a cohort study in China. Science China. Life sciences 2019, 62, doi:10.1007/s11427-018-9831-5.
  2. Wood, C.; Witt, W.; Skinner, A.; Yin, H.; Rothman, R.; Sanders, L.; Delamater, A.; Flower, K.; Kay, M.; Perrin, E. Effects of Breastfeeding, Formula Feeding, and Complementary Feeding on Rapid Weight Gain in the First Year of Life. Academic pediatrics 2020, doi:10.1016/j.acap.2020.09.009.
  3. Rzehak, P.; Sausenthaler, S.; Koletzko, S.; Bauer, C.; Schaaf, B.; von Berg, A.; Berdel, D.; Borte, M.; Herbarth, O.; Krämer, U., et al. Period-specific growth, overweight and modification by breastfeeding in the GINI and LISA birth cohorts up to age 6 years. European journal of epidemiology 2009, 24, doi:10.1007/s10654-009-9356-5.
  4. Bell, K.; Wagner, C.; Feldman, H.; Shypailo, R.; Belfort, M. Associations of infant feeding with trajectories of body composition and growth. The American journal of clinical nutrition 2017, 106, doi:10.3945/ajcn.116.151126.

Point 4: Results: Is there data about the type of formula and ratio of formula: breast milk in the mixed group? 

Response 4: Thanks for this comment. Indeed, we have monthly data of the type of feeding children received, which in children from group 2 (mixed feeding) is quite heterogeneous, given that most of them received both, human and formula milk from the very beginning, but some others had exclusive breastfeeding por 1 or 2 months.

Point 5: Discussion (line 282-285): Consider expanding the explanation about some of the mechanisms and clarify which mechanism may explain the study's outcome.  Also, reconsider the last sentence's inclusion of higher protein content of formulas as a behavior. Is there data about other feeding practices used by the formula group in the current study?

Response 5: Thanks for this comment. More detailed information has been included in lines 349 – 352.

The increased risk of RWG in formula fed infants may be due to a number of different mechanisms, like feeding to schedule versus feeding on demand [30] and the fact of being bottle-fed and not fed at breast [31], which may also have a positive long-term effect in terms of appetite regulation [32].“

References:

  1. Mihrshahi, S.; Battistutta, D.; Magarey, A.; Daniels, L. Determinants of Rapid Weight Gain During Infancy: Baseline Results From the NOURISH Randomised Controlled Trial. BMC pediatrics 2011, 11, doi:10.1186/1471-2431-11-99.
  2. Li, R.; Magadia, J.; Fein, S.B.; Grummer-Strawn, L.M. Risk of bottle-feeding for rapid weight gain during the first year of life. Arch Pediatr Adolesc Med 2012, 166, 431-436, doi:10.1001/archpediatrics.2011.1665.
  3. Disantis, K.; Collins, B.; Fisher, J.; Davey, A. Do infants fed directly from the breast have improved appetite regulation and slower growth during early childhood compared with infants fed from a bottle? The international journal of behavioral nutrition and physical activity 2011, 8, doi:10.1186/1479-5868-8-89.

Regarding the second comment, unfortunately, not, we do not know other feeding practices used by the formula group in the current study, so we have included this a potential limitation of the results in lines 402 – 404:

Also, information regarding specific feeding practices like feeding to schedule or feeding on demand and the addition of cereals to bottles was not included in the analyses and may be important to consider their effect on growth and weight gain.

Point 6: Discussion (starting line 305): Consider explaining catch-up growth, a compensatory mechanism, in low-birth infants and subsequent metabolic programming.

Response 6: This explanation has been included in the discussion section, in lines 369 – 371.

“Because of the potential influence of low birth weight on accelerated postnatal catch-up growth [40], which occurs typically in the first 24 months of postnatal life, in infants with low birth weight, birth weight was included as a covariate in all models.”

Reference:

  1. Singhal, A. Long-Term Adverse Effects of Early Growth Acceleration or Catch-Up Growth. Annals of nutrition & metabolism 2017, 70, doi:10.1159/000464302.

Reviewer 2 Report

Review: ”RWG, infant feeding practices and subsequent BMI trajectories. The Calina study”.

Very interesting and relevant study.

But there are some concerns and/or clarifications that need to be addressed before publication.

Material and method

Was the study solely observational or any type of intervention?

Unfortunately I cannot get access to ref. 12. But it does not look like a published protocol with details.

L80         contact to the families was during the first two weeks of life and then consent, is that right?

L83         children were examined at birth – in the study? But you did not have consent at this time? Please clarify about examinations at birth. Who did that?

L84         re-examination at 2 weeks, monthly (1,2,4,6 and 9 month) and yearly (1,2,3,4,5 and 6 years). Who did the examinations and what data were obtained from these examinations? When did you obtain data on growth, information about parents, nutrition? Did you use questionnaires?

Anthropometry (outcome):

L94         Primary outcomes was z-scores. That is a good thing in order to standardize. But how about delta z-scores from birth to certain time points – especially to 6 months where you evaluate RWG?

Rapid weight gain (exposure):

L107      RWG was from birth to 6 months. Why not 4 month? You obtain data on nutrition for 120 days = 4 months. Then it would make sense to evaluate RWG from birth to 4 months.

Type of infant feeding (mediator):

L11         Please clarify – is group 1 solely formula feeding from birth to 4 months of age? Is group 3 solely breastfeeding from birth to 4 months of age? How about group 2 – could that be breastfeeding for 1 months and then formula? Breastfeeding for 3 months and then formula?

Covariates:

L124      When was the face-to face interview?

Results:

I am very concerned about the differences between groups. lower GA and BW in the RWG group.

How many were small for gestational age?

How many were breastfed / formula fed in the low BW compared to normal BW group.

This is why you should only look at delta z-scores from birth to a pre defined time point – 4 month or 6 months – whatever makes sense according to nutrition!!

And then make new graphs with the same starting point at birth.

Discussion:

Please reflect on ”intervention” when you do an observational study on breastfeeding. Do the mothers try to continue with breastfeeding for a longer time when you keep asking them about breastfeeding? Do they report correct data on breastfeeding when you ask every months or maybe less?

Author Response

Response to Reviewer 2 Comments

Material and method

Point 1: Was the study solely observational or any type of intervention?

Response 1: The CALINA study was solely observational and no intervention was performed.

Point 2: Unfortunately, I cannot get access to ref. 12. But it does not look like a published protocol with details.

Response 2: Thanks for this comment. This reference has been changed for the correct one, which is a published book in which the protocol was published in 2009.

Olivares, J.; Rodríguez, G.; Samper, P. Valoración del crecimiento y la alimentación durante la lactancia y la primera infancia en atención primaria; Prensas Universitarias de Zaragoza. Zaragoza, Spain, 2009.

Point 3: L80  contact to the families was during the first two weeks of life and then consent, is that right?

Response 3: 9. Yes, this is correct. For better understanding, we have explained more details about enrollment in lines 110 – 118:

“After families accepted to participate in the study, data regarding prenatal factors and birth characteristics were obtained from both, mothers and their newborn children, from clinical histories and direct interview with the family. Perinatal information from children was obtained after enrollment and children were periodically re-examined in Primary Care Centers at 2 weeks, monthly (at 1, 2, 4, 6 and 9 months) and yearly (at 1, 2, 3, 4, 5, and 6 years of age), in every assessment, besides clinical evaluation of health indicators, growth indicators (weight and length or height) were measured by pediatricians and nurses that had previously been trained for consistency between measurements. Also, information regarding feeding practices was obtained in order to know what type of feeding children were receiving by each month.”

Point 4: L83         children were examined at birth – in the study? But you did not have consent at this time? Please clarify about examinations at birth. Who did that?

Response 4: Yes, families were contacted in the first pediatric evaluations and after having the signature for participating in the study, data regarding prenatal factors and birth were collected from the clinical history from mothers and their children. Children were re-examined in every well-child checkup.

Point 5: L84         re-examination at 2 weeks, monthly (1,2,4,6 and 9 month) and yearly (1,2,3,4,5 and 6 years). Who did the examinations and what data were obtained from these examinations? When did you obtain data on growth, information about parents, nutrition? Did you use questionnaires?

Response 5: Yes, this is correct. For better understanding, we have explained more details about enrollment in lines 110 – 118.

Anthropometry (outcome):

Point 6: L94         Primary outcomes was z-scores. That is a good thing in order to standardize. But how about delta z-scores from birth to certain time points – especially to 6 months where you evaluate RWG?

Response 6: This is something we didn´t consider at the beginning. We have included the mean of delta z-scores from birth to 6 months. We have included a delta z-scores in Table 1 to better quantify the changes from one point to another (0 to 6 months) in the RWG (1.34±0.58 in WAz) and in the non-RWG group (-0.38±0.76 in WAz).

Rapid weight gain (exposure):

Point 7: L107      RWG was from birth to 6 months. Why not 4 month? You obtain data on nutrition for 120 days = 4 months. Then it would make sense to evaluate RWG from birth to 4 months.

Response 7: Thanks for this comment. We agree with the idea that both, RWG and type of feeding should be analyzed at the same points in life. We evaluated RWG between birth and 6 months because current literature usually assesses this period of life, so we decided to establish this period for later likely comparisons. On the other hand, type of feeding has been assessed in the first 4 months of life because this is the time range when we are 100% sure that infants had not received nothing else but milk (breastmilk, formula or a combination of both). Nevertheless, we think is pertinent to assess RWG during the first 4 months, so that its development can be properly associated with the type of feeding received during that time. After receiving you feedback, we have analyzed the main outcomes of our analyses (repeated measures ANOVA, correlations and mediation analyses), and, we have found similar results in terms of significance.

Type of infant feeding (mediator):

Point 8: L11         Please clarify – is group 1 solely formula feeding from birth to 4 months of age? Is group 3 solely breastfeeding from birth to 4 months of age? How about group 2 – could that be breastfeeding for 1 months and then formula? Breastfeeding for 3 months and then formula?

Response 8: Part of this explanation was already written in lines 152 – 156. However, more details regarding “mixed feeding” have been included in lines 155 – 158:

“The second category, mixed feeding, included children that received both, formula and breast milk. This group included that received 1 or 2 months of exclusive breastfeeding and then a combination of both or those that received breast milk and formula since they were born.”

Covariates:

Point 9: L124      When was the face-to face interview?

Response 9: Details regarding measurements has been included in lines 110 – 118:

“After families accepted to participate in the study, data regarding prenatal factors and birth characteristics were obtained from both, mothers and their new-born children, from clinical histories and direct interview with the family. Perinatal information from children was obtained after enrolment and children were periodically re-examined in Primary Care Centres at 2 weeks, monthly (at 1, 2, 4, 6 and 9 months) and yearly (at 1, 2, 3, 4, 5, and 6 years of age), in every assessment, besides clinical evaluation of health indicators, growth indicators (weight and length or height) were measured by paediatricians and nurses that had previously been trained for consistency between measurements. Also, information regarding feeding practices was obtained in order to know what type of feeding children were receiving by each month.”

Results:

Point 10: I am very concerned about the differences between groups. lower GA and BW in the RWG group.

Response 10: Thanks for this observation. Indeed, there are significatively differences between groups, therefore adjustments for gestational and birth weight have been performed in both, the repeated measures ANOVA and in the mediation model, in order to address associations despite significant potential factors. We thought that including birthweight as covariate was pertinent and enough to consider its effect in relation to weight gain.

Point 11: How many were small for gestational age?

Response 11: This information was not included in the first version of the manuscript. Even though we have included birthweight as a covariate, we agree that categorizing children in terms of classification of weight according to gestational age is relevant. According to the cut-off point of < p10, a 26.6 % of infants with RWG were SGA vs a 5.3 % of infants without RWG (0-6 mo). This information could be included in Table 1.

Point 12: How many were breastfed / formula fed in the low BW compared to normal BW group.

Response 12: Thanks for this comment. Differences in type of feeding according to birthweight classification are the following

Low birth weight

Normal birth weight

Breastfeeding

9.4%

40.3%

Mixed feeding

28.2%

27.0%

Formula feeding

62.4%

32.7%

Differences across categories are significative, p=0.000

Point 13: This is why you should only look at delta z-scores from birth to a pre-defined time point – 4 month or 6 months – whatever makes sense according to nutrition!! And then make new graphs with the same starting point at birth.

Response 13: Given that results remain similar, regardless of the time period considered for rapid weight gain (0-6 or 0-4 months), if you agree, we would rather keep the analyses as they are.

Discussion:

Point 14: Please reflect on ”intervention” when you do an observational study on breastfeeding. Do the mothers try to continue with breastfeeding for a longer time when you keep asking them about breastfeeding? Do they report correct data on breastfeeding when you ask every month or maybe less?

Response 14: Thanks for this comment. The CALINA study aimed to study growth and type of feeding during the first years of life in infants from the region of Aragon in Spain. In no case the intention was an intervention planed in any specific group, besides the usual information that families receive in the periodical assessment in the pediatric health center promoting recommended feeding practices.

Reviewer 3 Report

Comments to the Authors:

Introduction: The introduction needs reorganizing and revising. The literature on prevalence of obesity does not need to be so lengthy. The purpose of the study is to examine the effect of breastfeeding and formula feeding on rapid weight gain in infants. Only one study (NOURISH) has been cited in the introduction to justify the purpose. While more studies may not have looked at long-term impact of feeding on weight status, there is literature on breastfeeding versus formula feeding and rapid weight gain in infants for shorter duration. Authors need to provide literature supporting and justifying the study goals. Additionally, the literature on prevalence of obesity can be reported more concisely so the authors can spend more time on infant feeding and growth trajectories.

Line 41: typo ‘Obesity does not only affect adults….’ ‘in fact, over 40 million children….’

Line 58: Specify the pediatric associations (at least for Spain or Europe).

Methods:

Sentences need edits.

Line 96: Child length up to age 2 years was measured….

Line 135: To test whether the trajectories of BAz, WAz and HAz differed according to based on the presence or not of RWG and according to the type of infant feeding,…….

Line 87: Did the authors check for differences between the sample included (n=740) and excluded (n=862), particularly maternal factors such as age, education etc.

Discussion:

Discussion needs some revising and reorganizing. The topic and the findings of this study are very interesting. Some changes could be made to discuss and draw inferences from the literature cited in relation to the interesting findings of this study. While the authors have done work comparing the findings with previous studies, focusing on the mechanism of growth trajectories and factors affecting them (included and not included in the current study) will make the discussion more interesting.

Line 283: Please specify the mechanisms responsible for rapid weight gain in formula fed infants. Authors have provided three citations for this sentence but it would help the readers if the reasons are discussed to convey the authors’ points across.

284: edit to say the “use of high protein formula” and “adding cereal into the bottle”, since it’s the behavior of caregivers that the authors are focus on. Additionally, authors need to provide more details on use of high protein formula and growth trajectory.

Line 304: According to the authors, what could those additional factors be and could those be included in the future studies to improve the rigor?

Line 305-314: In line the authors discuss the relationship between RWG and BMI at 6 months and thereafter. The authors then provide an explanation of how low-birth weight infants have differing androgen levels and insulin resistance leading to central fat deposition and weight gain in low birth weight infants. Then, in line 305 authors talk about the findings from their study that RWG infants had low birth weight but higher BMI at 6 months; the mechanism for which has already been explained in the above paragraph. The sequence needs some reorganization to better explain the tentative relationship between birthweight, RWG and BMI as supported by the findings of this study.

Author Response

Response to Reviewer 3 Comments

Point 1: Introduction: The introduction needs reorganizing and revising. The literature on prevalence of obesity does not need to be so lengthy. The purpose of the study is to examine the effect of breastfeeding and formula feeding on rapid weight gain in infants. Only one study (NOURISH) has been cited in the introduction to justify the purpose. While more studies may not have looked at long-term impact of feeding on weight status, there is literature on breastfeeding versus formula feeding and rapid weight gain in infants for shorter duration. Authors need to provide literature supporting and justifying the study goals. Additionally, the literature on prevalence of obesity can be reported more concisely so the authors can spend more time on infant feeding and growth trajectories.

Response 1: Many thanks for these suggestions. Some changes and improvements have been performed in the introduction. The information on the prevalence of obesity is no longer so extensive. Also, references of studies about the effects of feeding on weight status have been included in lines 67 – 73:

“In this sense, type of feeding is of relevance, given that formula feeding has been identified as a risk factor for RWG during the 6 months of life [13] and, in counterpart, breastfeeding has been associated with lower odds of rapid increase in weight when evaluated in different time periods during the first 24 months [14,15]. Moreover, not only isolated differences have been observed, but also differences in growth trajectories in terms of weight and BMI z scores between breastfed and formula-fed infants, being those that had been breastfed, the ones with lower WAZ and BAZ trajectories [16].”

Point 2: Line 41: typo ‘Obesity does not only affect adults….’ ‘in fact, over 40 million children….’

Response 2: Thanks for this observation. This typo has been corrected.

Point 3: Line 58: Specify the pediatric associations (at least for Spain or Europe).

Response 3: 2 references have been included in lines 61 - 63:

“Breastfeeding is considered by paediatric organizations such as the Italian [8] and the American [9] Associations of Paediatrics and the WHO [10] as the best method of nourishment for babies and infants during their first year of life.”

References

  1. Davanzo, R.R., C. Corsello, G. Position Statement on Breastfeeding from the Italian Pediatric Societies. Ital J Pediatr 2015, 41, doi:10.1186/s13052-015-0191-x.
  2. Gartner, L.; Morton, J.; Lawrence, R.; Naylor, A.; O'Hare, D.; Schanler, R.; Eidelman, A. Breastfeeding and the use of human milk. Pediatrics 2005, 115, doi:10.1542/peds.2004-2491.
  3. Organization, W.H. Indicators for assessing infant and young child feeding practices. Availabe online: https://www.who.int/nutrition/publications/iycf_indicators_for_peer_review.pdf (accessed on 28.09.2020).

Methods:

Point 4: Sentences need edits.

Response 4: Thanks for this comment. The paper has been reviewed for grammar and clarity, nevertheless no native speaker has revised it yet, so maybe there is still a need for improvements in grammar and clarity.

Point 5: Line 96: Child length up to age 2 years was measured….

Response 5: Thanks for this observation. This typo has been corrected.

Point 6: Line 135: To test whether the trajectories of BAz, WAz and HAz differed according to based on the presence or not of RWG and according to the type of infant feeding,…….

Response 6: Thanks for this correction. Changes have been performed.

Point 7: Line 87: Did the authors check for differences between the sample included (n=740) and excluded (n=862), particularly maternal factors such as age, education etc.

Response 7: We did not check for differences between the included sample and the excluded one. Some details have been included in lines 122 - 127.

“Differences between socioeconomic characteristics of the included and the excluded sample were analysed, and significant differences were observed in maternal age (32.3±5.01y in the included vs. 31.21±5.3 y in the excluded sample, p=0,000), origin (14.6% of immigrants in the included sample vs. 33.2%  in the excluded sample, p=0,000) and education (25.2% of low educated mothers in the included sample vs. 32.3% in the excluded sample and 39.8% of high educated mothers vs 32.8% in the excluded sample, p=0,002).”

Discussion:

Point 8: Discussion needs some revising and reorganizing. The topic and the findings of this study are very interesting. Some changes could be made to discuss and draw inferences from the literature cited in relation to the interesting findings of this study. While the authors have done work comparing the findings with previous studies, focusing on the mechanism of growth trajectories and factors affecting them (included and not included in the current study) will make the discussion more interesting.

Response 8: Thanks for this comment, we think is important to mention that other lifestyle factors may affect growth trajectories. This has been included in lines 408 – 410.

“Future studies should consider addressing other potential factors, such as dietary intake, physical activity and sleep, which are lifestyle factors that may also affect growth trajectories in addition to type of feeding during the first months of life.”

Point 9: Line 283: Please specify the mechanisms responsible for rapid weight gain in formula fed infants. Authors have provided three citations for this sentence, but it would help the readers if the reasons are discussed to convey the authors’ points across.

Response 9: Thanks for this comment. A wider explanation has been included in lines 349 – 352.

The increased risk of RWG in formula fed infants may be due to a number of different mechanisms, like feeding to schedule versus feeding on demand [30] and the fact of being bottle-fed and not fed at breast [31], which may also have a positive long-term effect in terms of appetite regulation [32]

References

  1. Mihrshahi, S.; Battistutta, D.; Magarey, A.; Daniels, L. Determinants of Rapid Weight Gain During Infancy: Baseline Results From the NOURISH Randomised Controlled Trial. BMC pediatrics 2011, 11, doi:10.1186/1471-2431-11-99.
  2. Li, R.; Magadia, J.; Fein, S.B.; Grummer-Strawn, L.M. Risk of bottle-feeding for rapid weight gain during the first year of life. Arch Pediatr Adolesc Med 2012, 166, 431-436, doi:10.1001/archpediatrics.2011.1665.
  3. Disantis, K.; Collins, B.; Fisher, J.; Davey, A. Do infants fed directly from the breast have improved appetite regulation and slower growth during early childhood compared with infants fed from a bottle? The international journal of behavioral nutrition and physical activity 2011, 8, doi:10.1186/1479-5868-8-89.

Point 10: 284: edit to say the “use of high protein formula” and “adding cereal into the bottle”, since it’s the behavior of caregivers that the authors are focus on. Additionally, authors need to provide more details on use of high protein formula and growth trajectory.

Response 10: Thanks for these corrections, changes have been performed.

Point 11: Line 304: According to the authors, what could those additional factors be and could those be included in the future studies to improve the rigor?

Response 11: Thanks for this comment. We have included the following information:

“We also adjusted for different maternal factors and birth weight finding a statistically significant result between RWG and BMI suggesting that there should be other factors that are playing a role apart. Beyond birth weight, gestational age at birth, parental education, parental BMI before pregnancy, ethnicity and maternal smoking during pregnancy, factors that may play an important role include dietary intake [40], eating behaviours [41], parental feeding practices [42] and physical activity levels [43]

References:

  1. Nguyen, A.; Jen, V.; Jaddoe, V.; Rivadeneira, F.; Jansen, P.; Ikram, M.; Voortman, T. Diet quality in early and mid-childhood in relation to trajectories of growth and body composition. Clinical nutrition (Edinburgh, Scotland) 2020, 39, doi:10.1016/j.clnu.2019.03.017.
  2. Abdella, H.; El Farssi, H.; Broom, D.; Hadden, D.; Dalton, C. Eating Behaviours and Food Cravings; Influence of Age, Sex, BMI and FTO Genotype. Nutrients 2019, 11, doi:10.3390/nu11020377.
  3. Vaughn, A.E.; Ward, D.S.; Fisher, J.O.; Faith, M.S.; Hughes, S.O.; Kremers, S.P.; Musher-Eizenman, D.R.; O’Connor, T.M.; Patrick, H.; Power, T.G. Fundamental constructs in food parenting practices: a content map to guide future research. Nutr Rev 2016, 74, 98-117, doi:10.1093/nutrit/nuv061.
  4. Gao, Z. Growth Trajectories of Young Children's Objectively Determined Physical Activity, Sedentary Behavior, and Body Mass Index. Childhood obesity (Print) 2018, 14, doi:10.1089/chi.2018.0042.

Point 12: Line 305-314: In line the authors discuss the relationship between RWG and BMI at 6 months and thereafter. The authors then provide an explanation of how low-birth weight infants have differing androgen levels and insulin resistance leading to central fat deposition and weight gain in low birth weight infants. Then, in line 305 authors talk about the findings from their study that RWG infants had low birth weight but higher BMI at 6 months; the mechanism for which has already been explained in the above paragraph. The sequence needs some reorganization to better explain the tentative relationship between birthweight, RWG and BMI as supported by the findings of this study.

Response 12: Thank you. We have reorganized the main ideas of the discussion; we hope it es clearer now.

Round 2

Reviewer 2 Report

Thanks for letting med see the revised manuscript.

The manuscript has improved.

I still think it is wrong not to use delta z-scores for comparison on growth. Just comparing z-scores do not take into account about a different set point at birth. And you defnitely have two very different groups for comparison. A RWG yes group with af lower birth weigth and 65% SGA compared to a RWG not group with a higher birth weight and only 33% SGA. You are then comparing pears and apples (please find the publication from Embleton N on comparing apples and pears).

Please specify - also in the conclusion - that you are aware of comparing very different groups of infants. 

And at least please reflect on these very different groups. Different growth pattern among SGA compared to non SGA infants instead of feeding practice?

The reflection on intervention when asking questions every months has been described in other papers (e.g. Bruun S et al on sms questioning). Recall bias.

Author Response

Best regards.

Reviewer 3 Report

Dear Authors,

Thank you for incorporating the suggested revisions to the manuscript. The revised version offers more clarity to both introduction and discussion section. There are few minor editing changes that are required to improve grammar. Additionally, it strongly recommended to have the paper proof read for typos and grammatical errors.

  1. Line 63: Moreover, not only isolated differences have been observed, but also differences in growth trajectories in terms of weight and BMI z scores between breastfed and formula-fed infants, being those that had been with breastfed, the ones with lower WAZ and BAZ trajectories [16]. This line needs revision. Here is a suggestion

In addition to isolated differences, there have been differences in growth trajectories in terms of weight and BMI Z scores between breastfed and formula fed infants, with breastfed infants having lower WAZ and BAZ trajectories [16].

  1. Line 271: There is a typo – growth patterns is repeated.

Line 301: The increased risk of RWG in formula fed infants may be due to a number of different mechanisms, like feeding to schedule versus feeding on demand [30] and the fact of being bottle-fed and not fed at breast [31], which may also have a positive long term effect in terms of appetite regulation [32]. This sentence needs revision and clarity. I would recommend breaking down the thoughts in more than one sentence. The fact of being bottle fed and not breast fed - needs clarity as all formula fed infants are bottle fed. I think the authors are trying to point out that infants feeding at breast may have a better appetite regulation. Please revise for clarity. Secondly, these are “factors” rather than “mechanisms”.

  1. Line 311: Likewise, the results of our study the delay of bottle-feeding introduction had a protective effect against obesity at 6 years of age in a sample of Spanish children. These results highlight the need for greater support of breastfeeding to avoid future childhood obesity [34,35].

This sentence needs revision. “Likewise, the results of our study…..” it seems authors are discussing findings of their study when findings of a different study are discussed. Suggestion to revise – Similarly, delay of bottle feeding……….in Spanish children, highlighting the need for greater support…….childhood obesity [34,35].

  1. Line 336: there is typo – significant

Author Response

Best regards.
